# AGSAM: Agent-Guided Segment Anything Model for Automatic Segmentation in Few-Shot Scenarios

**DOI:** 10.3390/bioengineering11050447

**Published:** 2024-04-30

**Authors:** Hao Zhou, Yao He, Xiaoxiao Cui, Zhi Xie

**Affiliations:** 1State Key Laboratory of Ophthalmology, Guangdong Provincial Key Laboratory of Ophthalmology and Visual Science, Zhongshan Ophthalmic Center, Sun Yat-Sen University, Guangzhou 510000, China; wuwuwu212414@live.com (H.Z.); scheyao@hotmail.com (Y.H.); 2Joint SDU-NTU Centre for Artificial Intelligence Research (C-FAIR), Shandong University, Jinan 250000, China

**Keywords:** deep learning, medical image segmentation, cardiac ultrasound, Segment Anything Model, few-shot scenarios

## Abstract

Precise medical image segmentation of regions of interest (ROIs) is crucial for accurate disease diagnosis and progression assessment. However, acquiring high-quality annotated data at the pixel level poses a significant challenge due to the resource-intensive nature of this process. This scarcity of high-quality annotated data results in few-shot scenarios, which are highly prevalent in clinical applications. To address this obstacle, this paper introduces Agent-Guided SAM (AGSAM), an innovative approach that transforms the Segment Anything Model (SAM) into a fully automated segmentation method by automating prompt generation. Capitalizing on the pre-trained feature extraction and decoding capabilities of SAM-Med2D, AGSAM circumvents the need for manual prompt engineering, ensuring adaptability across diverse segmentation methods. Furthermore, the proposed feature augmentation convolution module (FACM) enhances model accuracy by promoting stable feature representations. Experimental evaluations demonstrate AGSAM’s consistent superiority over other methods across various metrics. These findings highlight AGSAM’s efficacy in tackling the challenges associated with limited annotated data while achieving high-quality medical image segmentation.

## 1. Introduction

AI-based automated techniques have demonstrated their efficiency and success in medical image segmentation [1,2], which is indispensable for disease diagnosis and progression assessment. However, the foundation of high-quality AI segmentation models lies in obtaining abundant high-quality annotated samples [3]. However, segmentation annotation is difficult and laborious for clinical applications, posing a challenge to effectively harnessing limited annotation [4], compared to other medical tasks such as classification [5,6].

This challenge of data scarcity can be addressed through few-shot learning approaches [7]. Few-shot learning enables pre-trained models to generalize and segment new categories of data that were unseen during the original training by leveraging just a few labeled samples per class.

The recent success of Generative Pre-Trained Transformer (GPT) models in few-shot tasks [8,9,10], pre-trained on vast datasets, has inspired the development of the Segment Anything Model (SAM), a model trained extensively on data to encode and decode feature for segmentation, exhibiting remarkable few-shot and even zero-shot capabilities [11]. SAM-Med2D [12] bridges the gap between SAM’s proficiency in natural images and its application in medical 2D image analysis. Studies on SAM and SAM-Med2D have revealed that simple prompts, such as a few point coordinates or rough region vertices, can effectively guide precise segmentation [12]. However, these methods still rely on additional manual prompts, limiting their applicability.

Recent work has leveraged SAM’s pre-trained image encoder for automated segmentation of medical images, like AutoSAM [13] and nnSAM [4], by integrating it into new architectures. However, these approaches focus only on SAM’s encoding module while underutilizing its extensively trained decoding module. SAM’s decoder holds strong value in few-shot scenarios where limited training data make fine-tuning a new decoder inferior to utilizing SAM’s already optimized decoder. Moreover, while SAM-Med2D fine-tunes SAM on medical images, it was reported that updating both SAM’s encoder and decoder modules, either individually or jointly, enhances segmentation performance, with joint updates achieving greater gains. This suggests that after extensive pre-training on abundant data, SAM’s encoder and decoder inherently acquire valuable knowledge beneficial for segmentation tasks [12]. Hence, fully utilizing both the encoding and decoding capabilities within a trained SAM represents an unmet need for maximizing performance, especially with scarce training data.

In the few-shot scenario, data are exceedingly scarce, rendering them inadequate for direct fine-tuning of the pre-trained SAM. Consequently, the paper describes training a lightweight segmentation model, known as the agent model, using these limited data. The predictions and acquired semantic features of this agent model are subsequently leveraged to generate prompts for the pre-trained model, thereby activating its encoding and decoding capabilities. Therefore, we propose a simplified agent-guided model to replace manual prompt generation, introducing an optimized framework called Agent-Guided SAM (AGSAM). AGSAM maximizes the utilization of SAM’s trained encoder and decoder modules. The fusion of SAM and the guided model is particularly advantageous for limited training data, achieving high-quality medical image segmentation. The main contributions of this paper are summarized as follows:Introducing a novel fully automatic segmentation approach: AGSAM. This method leverages SAM-Med2D’s pre-trained feature extraction and generalized decoding capabilities. The guided model seamlessly replaces prompt generation and embedding, making it adaptable to any segmentation method.Introduction of a feature augmentation convolution module (FACM) in AGSAM, a parameter-free and computationally efficient module that enhances model accuracy. FACM ensures more stable feature representations, minimizing image noise impact on segmentation.Experimental comparisons demonstrate that AGSAM consistently outperforms comparative methods across various metrics, showcasing its effectiveness in few-shot scenarios.

## 2. Related Work

### 2.1. SAM and SAM-Med2D

The success of large language models like GPT series models in zero-shot and few-shot scenarios when trained on extensive datasets has translated to the vision domain through SAM [11]. Trained on a massive dataset of 11 million images and over a billion segmentation masks, SAM has emerged as a pivotal advancement for segmentation tasks, exhibiting significant potential for few-shot or zero-shot learning across diverse image categories. However, directly applying the pre-trained SAM to medical images reveals limitations in accuracy without additional domain-specific fine-tuning [12].

Thus, SAM-Med2D extended SAM’s framework by introducing an adapter mechanism to acquire domain-specific knowledge from a large dataset of approximately 4.6 million medical images across diverse medical devices and 197,000 segmentation masks. This establishes a robust foundation tailored for medical image segmentation applications.

### 2.2. SAM-Based Method

Building upon SAM, a series of innovative methods have been developed to further enhance its capabilities.

AutoSAM goes beyond SAM’s feature encoding by integrating an external lightweight task prediction head, aiming to leverage SAM’s strong image encoding while extending applicability to a broader range of medical segmentation tasks for fully automated operation [13]. nnSAM serves as a plug-and-play solution that merges SAM’s embedded features with encoded features from other segmentation models, executing predictions through the other model’s decoding section [4]. This approach leverages SAM’s encoding capabilities while integrating into established segmentation architectures. However, these existing SAM-based methods primarily focus on utilizing SAM’s encoding module. As depicted in Figure 1, they essentially embed SAM’s encoding into specific models rather than enhancing and optimizing SAM’s complete workflow framework. Notably, none of these methods leverage the domain-specific pre-training of SAM-Med2D on massive medical image datasets. Inspired by SAM’s strong performance, our objective is to unlock the potential of SAM’s comprehensive set of modules, including its extensively trained decoder, for few-shot medical image segmentation by taking full advantage of the SAM-Med2D pre-training on medical data.

## 3. Methods

### 3.1. Architecture Overview

The existing methodologies leverage SAM’s encoded features directly by integrating various segmentation prediction heads for automated segmentation, effectively utilizing SAM’s encoding component to support other models (Figure 1). In contrast, our proposed approach integrates the pre-trained model’s encoder and decode module with the agent model to guide or assist the SAM. The AGSAM framework introduces a strategy by integrating an additional segmentation model as a guiding agent within the SAM architecture (Figure 2). AGSAM’s primary objective is to fully exploit SAM’s pre-trained image-feature-encoding and prompt- and mask-decoding capabilities to enable autonomous segmentation without relying on manual prompts. By coupling SAM with the guiding segmentation model, AGSAM automates the prompt generation process typically required for SAM, accurately performing segmentation without human intervention or manual prompts.

### 3.2. Agent-Guided SAM

The AGSAM architecture consists of two primary modules: the agent model and the pre-trained SAM-Med2D (Figure 2). When an image is input into AGSAM, its features are extracted separately by the agent model and SAM’s image feature encoder. This dual-path feature-embedding process enhances the diversity of features.

In this process, the input image I, after data pre-processing, is transformed into a tensor with dimensions of 3 × 256 × 256. First, for the feature pathway in SAM, the feature $f_{SAM}$ was computed through the pre-trained encoder module $M_{SAM_{encoder}}$. Specifically, the image is tokenized by partitioning it, followed by feature extraction and encoding through 12 layers of transformer modules within the Vit-b structure, ultimately resulting in a feature of SAM $f_{SAM}$ with dimensions of 256×16×16.
(1)$$f_{SAM}=M_{SAM_{encoder}}\left(I\right)$$

Additionally, the other pathway is determined based on the selection of the agent model. Taking FCN as an example, for the same input image I, the agent encoder $M_{agent_{encoder}}$ is the feature extract module of FCN. It can encode the input to obtain the feature of the agent $f_{Agent}$ with dimensions of 2048×32×32.
(2)$$f_{Agent}=M_{agent_{encoder}}\left(I\right)$$

After obtaining the two features, they are fused by confusion module $M_{CM}$ as follows: $f_{SAM}$ is aligned with $f_{Agent}$ in terms of channels and spatial dimensions through a 3 × 3 convolutional layer ${Conv}_{3\times 3}$ and a linear interpolation operation $F_{bilinear}$, resulting in $f_{SAM}^{\prime }$. Then, they are concatenated along the channel dimension. Finally, a 1 × 1 convolutional layer ${Conv}_{1\times 1}$ is applied to reduce the dimensionality of the concatenated feature channels. The entire process is represented as:(3)$$f_{Fused}=M_{CM}\left(f_{Agent}\;,{\;f}_{SAM}^{\prime }\right)={Conv}_{1\times 1}(f_{Agent}⊕F_{bilinear}\left({Conv}_{3\times 3}\left(f_{SAM}\right)\right))$$

After obtaining $f_{Fused}$, the agent model utilizes the agent decoder $M_{agent_{decoder}}$ based on this information to decode and obtain the segmentation result ${Mask}_{agent}$. Taking FCN [14] as an example, this agent decoder is the mask prediction module in FCN.
(4)$${Mask}_{agent}=M_{agent_{decoder}}\left(f_{Fused}\right)$$

Then, the prompt generation module employs the prediction of agent results and confused feature for embedding sparse and dense prompt features, respectively. There are two separate paths for generating sparse prompt embedding and dense prompt embedding (Appendix A). Firstly, for sparse prompt embedding $p_{sparse}$, the obtained prediction of agent ${Mask}_{agent}$ undergoes downsampling compression to 1/16 of its original size through pooling layer Avgpool and three FACMs with a stride of 2, FACMs (Appendix A). Subsequently, it goes through four FACMs, FACMs, for feature enhancement while maintaining the feature map size. Then, a single 1 × 1 convolutional layer ${Conv}_{1\times 1}$ is applied to adjust the channel dimensions of the feature map. The channel dimension is updated to *n* × 256, where *n* represents the number of segmentation categories. Finally, the tensor is arranged to obtain parse prompt embedding $p_{sparse}$.
(5)$$p_{sparse}=Arrange\left({Conv}_{1\times 1}\left(\left(FACMs\left(Avgpool\left({Mask}_{agent}\right)\right)\right)\right)\right)$$

For dense prompt embedding $p_{dense}$, the entire process is similar to sparse prompt embedding, except that the input is the fused feature $f_{fused}$ instead of the mask ${Mask}_{agent}$. Firstly, $f_{fused}$ is resized through interpolation to match the feature map size of ${Mask}_{agent}$, resulting in $f_{fused}^{\prime }$. Then, it undergoes a similar process as before for calculation. Finally, after arranging, dense prompt embedding $p_{dense}$ can be obtained.
(6)$$p_{dense}=Arrange\left({Conv}_{1\times 1}\left(\left(FACMs\left(Avgpool\left(f_{fused}^{\prime }\right)\right)\right)\right)\right)$$

After obtaining the two prompt-embedding features, the pre-trained SAM’s mask decoder module $M_{{SAM}_{decoder}}$ is activated to generate the predicted result ${Mask}_{SAM}$. At the same time, the prediction of SAM is fused with the prediction of the agent to obtain the fused prediction result ${Mask}_{pred}$:(7)$${Mask}_{SAM}=M_{{SAM}_{decoder}}\left(p_{sparse},\;p_{dense}\right)$$
(8)$${Mask}_{pred}=(1-\alpha ){\times Mask}_{SAM}+{\alpha \times Mask}_{agent}$$where a ϵ [0.0, 1.0]. In the study, several empirical values are chosen, such as 0.1, 0.3, 0.5. The overall loss $L_{all}$ during the training of AGSAM is the sum of two parts: the loss $L_{Agent}$ constraint between the prediction of agent ${Mask}_{agent}$ and the annotation, and the loss $L_{pred}$ constraint between the final fused mask ${Mask}_{pred}$ and the same annotation. It can be represented as:(9)$$L_{all}=L_{Agent}+L_{pred}$$

### 3.3. Feature Augmentation Convolution Module

To further enhance the model’s generalization ability in few-shot scenarios, mitigate the risk of overfitting, and augment feature responses, the feature augmentation convolution module (FACM) was introduced. This module introduces a feature enhancement mechanism through random linear suppression, expressed as:(10)$$x_{aug}=\alpha \times x+\beta$$

Here, α follows a range of $\left[0.25,1.0\right]$, and β ranges from [−10,10]. The FACM integrates random linear suppression to smooth and adjust feature information, selectively suppressing less discernible features. The OFF state of FACM status is equivalent to a single convolutional layer. During training phase, the state of FACM can be either ON/OFF and is OFF during the inference phase. This mechanism guides the model toward more contrasted response feature values, resulting in an augmented feature set that enriches the overall information content. This allows for feature response enhancement without adding extra parameters or increasing computational complexity, expressed as:(11)$$x^{\prime }=M_{FCAM}\left(x\right)=\left\{\begin{array}{c} {a\times Conv}_{3\times 3}\left(x\right)+\beta +{Conv}_{3\times 3}\left(x\right),status:ON\;\\ {1\times Conv}_{3\times 3}\left(x\right)+0+{Conv}_{3\times 3}\left(x\right),\;status:OFF \end{array}\right.$$

## 4. Experiments

### 4.1. Datasets

The CAMUS [15] dataset comprises cardiac ultrasound images from 500 patients in apical two-chamber (A2C) and apical four-chamber (A4C) views. Collected from a single vendor (GE Vingmed Ultrasound, Horten Norway) and center (Saint-Etienne University Hospital, France), it represents a highly heterogeneous dataset encompassing varying image quality and pathological cases, representing typical real-world clinical practice data. For each patient, CAMUS provides complete cardiac cycles with manual annotations delineating the end-diastolic (ED) and end-systolic (ES) cardiac structures for each view. As annotations were unavailable for the last 50 patients in the training data, the remaining 450 patients were utilized for training and testing purposes. With four different orientations imaged per patient, the dataset contains a total of 1800 annotated images. The dataset was split into 512 samples for the training set and 88 samples for the validation set, with the remaining 1200 samples used for testing. The annotations include the endocardium and epicardium of the left ventricle, as well as the left atrium wall, serving as the ground truth for both supervised training and evaluation. Accurate segmentation of these regions is crucial for subsequent clinical measurements such as ventricular volume and ejection fraction.

The REFUGE [16] dataset consists of 1200 retinal color fundus photographs (CFPs) acquired from patients in an upright sitting position using one of two devices: a Zeiss Visucam 500 or a Canon CR-2. Manual annotations of the optic disc (OD) and optic cup (OC) boundaries were provided by seven independent glaucoma specialists, serving as the ground truth for both supervised training and evaluation. The first 100 samples (50 for train and 50 for validation) were utilized for training and validation to select the optimal model weights. Subsequently, the remaining 1100 samples were used for testing purposes. Accurate segmentation of the OD and OC regions enables assessment of a patient’s risk of glaucoma based on the size ratio between these structures.

### 4.2. Data Pre-Processing

For the CAMUS dataset, the data pre-processing includes converting grayscale images to the 3-channel RGB images, transforming pixel values to integer values ranging from 0 to 255. During training, both images and annotated masks undergo random central rotations of ±10°. Finally, the images are uniformly resized to 256 × 256 pixels for training.

For the REFUGE dataset, as retinal fundus photographs are inherently RGB three-channel images, there is no need for converting single-channel to multi-channel data. The remaining pre-processing steps are consistent with those used for the CAMUS dataset.

### 4.3. Implementation Details and Evaluation Metrics

#### 4.3.1. Architecture Description

Our AGSAM builds upon the SAM-Med2D method, which employs SAM’s Vit-b structure consisting of 12 transformer layers, each with 12 attention heads. The encoding dimension in each layer is set to 768, followed by two convolutional layers to reduce the feature embedding to 256, producing the SAM feature. On the other side, the guiding model is a basic segmentation model with an encoder to decoder structure, taking FCN as an example. Its encoder is the standard ResNet-50 structure before global pooling, resulting in a feature of the agent with dimensions $\frac{h}{8}\times \frac{w}{8}\times 2048$, where h and w are the height and width of the input image, respectively.

In the fusion module, the feature of SAM was aligned with the feature of the agent through a single convolutional layer, followed by interpolation to match the size of the agent’s feature and concatenation along channels. A 1 × 1 convolution layer is then applied for channel fusion, compressing it by half. The fused feature undergoes encoding embedding and mask prediction in the guiding model’s decoder. Using FCN as an example, the decoder consists of two convolutional operations to adjust the channel dimensions to the number of segmentation classes, followed by linear interpolation upsampling operation to obtain the predicted mask for the agent model.

The predicted mask and the fused feature are input into the prompt generation module. This module contains two channels, each processing one of the two feature types. The two pathways share a similar structure, involving average pooling for feature downsampling. Seven layers of FACMs are utilized for further feature extraction, with the final 1 × 1 convolution reshaping the channels into 768. Notably, within the seven FACM blocks, only the first layer activates the augmentation block (ON), while the remaining layers do not (OFF), and the convolution blocks in the first three FACMs have a stride of 2, while the subsequent four layers have a stride of 1 (Appendix A). The output represents the feature embeddings for the two types of prompts, corresponding to sparse prompts generated from the predicted mask and dense prompts generated from the fused feature.

The FACM within the prompt generation module consists of a convolutional block and an augmentation block employing random linear transformations. During training and when the status is set ON, the augmentation block is activated with a certain probability, suppressing the response values of the convolutional feature map or setting the feature response values to zero with small probability. During inference or when its status is set OFF, it behaves like a standard 3 × 3 convolutional block.

Finally, the fused image feature embedding, along with the generated feature embeddings for two types of prompts, is fed into the mask decoder of SAM. This module directly adopts SAM’s flood-decoding module, essentially combining the image feature embedding and various prompt feature embeddings and producing the predicted mask results. The ultimate mask prediction result is a weighted combination of the mask result and the guiding model’s predicted mask.

#### 4.3.2. Setting of Training

All methods were implemented using PyTorch, and both training and testing were conducted on a single NVIDIA GPU (RTX 4090 with 24 GB memory). The training process for all methods involved initially augmenting the data to 512 copies, followed by training for 50 epochs. The batch size was set to 8, except nnSAM which used 4 and Mamba-Unet which used 1. Dice loss was selected as the loss function. The optimizer was AdamW, with an initial learning rate of 1 × 10^−4^ (except SegFormer was set 1 × 10^−5^). Starting from the 26th epoch, the learning rate decayed to 1 × 10^−5^ (for SegFormer was set 1 × 10^−6^). The best model for evaluation was selected by the performance on the validation set.

#### 4.3.3. Evaluation Metrics

For evaluation, we employed the dice similarity coefficient (DICE) and Hausdorff distance (HD) as metrics. The DICE assessed the similarity of segmentation results, representing internal overlap [17], while HD focused on the similarity of contour portions in the segmentation results [18]. These metrics systematically evaluated the segmentation effectiveness, considering both internal and contour regions. Additionally, we employed additional metrics such as sensitivity, specificity, AUC, and AUPR in the Appendix A for further analysis.

### 4.4. Experimental Settings

To assess AGSAM’s performance, we conducted various experiments for comparisons and an ablation study. To ensure fairness in the comparisons, all methods considered in the study were systematically tested by reproducing their core functionalities and utilizing open-source code. Training or fine-tuning was uniformly performed on the research dataset. More details about hyperparameters of experiments and training can be seen in Appendix A. A *p* < 0.05 was set to determine significance, and *p* values were two-sided (*t*-test).

#### 4.4.1. Comparative Study

We conducted comparisons in CAMUS and REFUGE datasets between various recent baseline segmentation models and optimized baseline models based on SAM. The baseline segmentation models include both traditional CNN models and newer transformer-based state-of-the-art solutions, specifically FCN [14], DeepLabV3 [19], PSPNet [20], Fast-SCNN [21], TGANet [22], SegFormer [23], Unet++ [24], and Mamba-Unet [25]. Additionally, we compared the latest optimized baseline solutions based on SAM, such as autoSAM [13] and nnSAM [4]. Our comprehensive comparison now encompasses representative state-of-the-art solutions across different architectures: pure CNN-based (FCN, DeepLabV3, PSPNet), transformer-based (SegFormer, SAM-Med2D), hybrid CNN–transformer models (nnSAM, autoSAM), and the newly added Mamba-Unet. Since our evaluation focused on automated segmentation, SAM [11], SAM-Med2D [12], and MedSAM [26], which require manual prompts, were not included in the comparison.

For the baseline segmentation models, we utilized publicly available and widely used source codes. However, in the case of nnSAM, due to our hardware constraints and the unique configuration of its core network, nnUnet, which differs from other methods evaluated in our study, we followed the description provided in its paper to reproduce the nnSAM method. During this process, we replaced the core network with a lighter network solution and substituted MobileSAM [27] with SAM-Med2D.

#### 4.4.2. Ablation Study

We conducted ablation experiments to validate the effectiveness of integrative modules with one training sample. We used FCN, DeepLabV3, and Unet++ as three segmentation baselines to evaluate the performance of different combinations of the image feature encoder of SAM (FE), mask decoder of SAM (MD), and feature augmentation convolution module (FACM). To verify the effectiveness of AGSAM’s strategies, we performed ablations on the structure of the agent model, comparing the impact of different agent model structures between nnSAM and our proposed method. Specifically, based on the proposed method and nnSAM, we used FCN (nnSAM (FCN) vs. proposed (FCN)) and DeepLabV3 (nnSAM (Deep) vs. proposed (Deep)) to compare the two methods. All results were evaluated using DICE and Hausdorff distance (HD) metrics to assess differences in the impact on segmentation region internal and edge regions. Additionally, we included a series of supplementary evaluations to validate the methods proposed in the paper. These evaluations encompassed the computational efficiency of different methods, assessing the influence of additional data augmentation on the results, and comparing the performance of the automatically prompted scheme proposed in the paper with the manually provided coordinate prompting information of the SAM-Med2D method.

## 5. Results and Discussion

### 5.1. Comparison in Few-Shot Scenario with CAMUS

We evaluated each method’s performance across varying amounts of training data. As expected, the evaluation metrics showed consistent improvement for all methods as the data volume increased (Table 1, Appendix A). Notably, AGSAM consistently outperformed the other approaches, underscoring the effectiveness of fully leveraging the pre-trained SAM. However, this performance differential gradually narrowed with increasing training data volume. This can be attributed to the other approaches having more data available to better fine-tune their models when provided with larger datasets. AGSAM’s advantage was most pronounced in limited data scenarios, where its ability to exploit the pre-trained SAM provided significant benefits over methods that rely heavily on data-driven fine-tuning from scratch.

As the data volume increased to a more typical few-shot problem, the adoption of pre-trained weight initialization methods (ResNet-50 pre-trained with ImageNet dataset) such as FCN, DeepLabV3, and PSPNet demonstrated significant superiority over random initialization, as evidenced by the performance comparison with Unet++ (Figure 3 and Appendix A). Comparing DeepLabV3 with AGSAM, noteworthy improvements by AGSAM were observed in all categories except for Endocardium. For example, Epicardium’s result was enhanced from 0.4248 to 0.5372, and that of the Left Atrium Wall from 0.4112 to 0.5073, while the Endocardium metric experienced a slight decrease from 0.7112 to 0.6829 (Appendix A). This can be attributed to the Endocardium’s prevalent central positioning and near-uniform elliptical shape within the images, allowing methods like DeepLabV3 to more effectively learn relevant features even with extremely limited data. As data volume increased, AGSAM outperformed various baselines in achieving more accurate segmentations for all three structures (Appendix A). Ultimately, when trained with 512 samples, despite metrics being close across methods, AGSAM still yielded the best overall results (Appendix A).

Visual examination of the predicted segmentation results revealed distinctive characteristics (Figure 3). AGSAM exhibited fewer false positive segmentation pixels, especially in limited training data scenarios. This was particularly noticeable for the Left Atrium Wall segmentations, which were poor across all methods until more training data were provided. However, even with increased data, SegFormer continued to struggle with reliable predictions for this category. These observations highlight AGSAM’s efficient data utilization in few-shot scenarios and more generalized feature extraction and semantic induction capabilities.

### 5.2. Comparison in Few-Shot Scenario with REFUGE

Multiple experiments were conducted on the REFUGE dataset using varying data volumes, following comparative studies for the CAMUS dataset. AGSAM consistently outperformed other solutions in most cases across different evaluation metrics. However, as the dataset volume increased, the performance advantage of the proposed method gradually diminished, aligning with the trends observed in previous experiments (Table 2). Additional indicators and detailed category-wise results are provided in the Appendix A, with AGSAM performing the best for both the optic cup and optic disc (Appendix A).

Visual inspection of the predicted segmentation results revealed that even with very limited data, AGSAM exhibited higher sensitivity in detecting the optic cup area within the images (Figure 4). As more data became available, all methods demonstrated noticeable segmentation improvements. However, methods like nnSAM, Mamba-Unet, TGANet, and SegFormer exhibited significant false positives under a few data volumes. In contrast, the results of AGSAM were very stable, with the predicted areas gradually aligning more closely with the annotated ground truth.

### 5.3. Ablation Studies of AGSAM

The ablation experiments were conducted with training data of 1 for the CAMUS dataset. At first, the baseline model FCN was tested as the agent model. Successively, SAM feature encoder (FE), which corresponds to nnSAM (FCN), SAM’s mask decoder (MD), and the feature augmentation convolution module (FACM) were incrementally added. Metrics revealed improvements with the addition of each module over the baseline. Finally, the combination of all modules resulted in the maximum enhancement from 0.4819 to 0.5419 (Table 3). Metrics demonstrated improvements with each module over the baseline, and the combination of all modules resulted in the maximum enhancement from 0.4819 to 0.5419. Furthermore, we assessed FCN, DeepLabV3, and Unet++ as guided models for AGSAM. Consistent trends observed across methods (e.g., for DeepLabV3 from 0.5157 to 0.5758; for Unet++ from 0.2486 to 0.2615) (Table 3) suggest the effectiveness of AGSAM’s modules, rather than reflecting inherent randomness associated with a single model.

To assess the efficacy of FACM feature augmentation, we visualized the extracted feature maps [5,6] (Figure 5) using Grad-CAM [28]. This visualization technique revealed that after FACM processing, the feature maps exhibited improved alignment and brightness with annotated regions, indicating a better correspondence between the augmented features and the ground truth annotations. This observation suggests that suppressing irrelevant feature responses during training enhances the model’s ability to produce relevant responses.

The design intention behind the FACM was to ensure that during supervised training, the feature maps’ response values and positions at each layer in the current iteration contain semantic feature information contributing to the final segmentation result. By randomly compressing the values of the feature maps, the model is encouraged to prioritize larger response values in subsequent iterations to counteract the effects of random compression. Additionally, the FACM aims to minimize the inclusion of feature response regions irrelevant to the final prediction.

The visualization results (Figure 5) demonstrate that the integration of the FACM effectively suppresses irrelevant feature information while enhancing feature responses relevant to the prediction results. This enhancement allows the prompt feature embedding to extract relatively reliable information from noisy or rough data, contributing to improved segmentation performance.

Although the guided model may produce unreliable prediction information, leading to unreliable prompt information, the impact of inaccuracies can be mitigated in two ways: (1) the subsequent prompt-embedding module, which can accommodate certain noise information, coupled with the immunity of SAM’s decoder to a certain extent of erroneous cues, and the final decoding results correct some of the prediction errors made by the agents used to generate prompt embeddings. (2) The integration of the FACM can suppress some irrelevant feature information and enhance feature responses related to the prediction results, as shown in the figure, enabling the prompt feature embedding to extract relatively reliable information from rough data.

AGSAM predictions are a weighted sum of the agent and SAM decoder (Table 4). Varying weights showed higher decoder weights are crucial with limited data, relying on SAM’s generalized decoding. As data increase, the agent gains more task experience through training. Consequently, the agent’s prediction weights can be gradually strengthened. We believe specialized experience will surpass generalized experience given sufficient data. Fine-tuning the decoder could then further improve accuracy.

In terms of computational efficiency comparison, although the proposed method has a larger computational load due to the presence of the SAM, it can still process predictions for 35 samples per second, meeting the requirements for real-time processing (Appendix A). Evaluation of the proposed method using FACMs in two different states (ON/OFF) showed no increase in processing time, indicating that FACMs do not add computational overhead.

For the data augmentation experiment, random displacement and random scale scaling were added to the original settings (Appendix A). Comparing the results with the previous settings, data augmentation was found to increase the diversity of training data through simulated data, thereby alleviating the scarcity of diversity in few-shot scenarios (Appendix A).

Compared with SAM-Med2D augmented with manual prompts, AGSAM can achieve better results in a fully automatic manner compared to manual prompts (Appendix A). AGSAM may provide a solution to enhance the performance of SAM-Med2D for specific medical segmentation applications.

### 5.4. Limitations of AGSAM

However, our current study has some limitations that should be addressed in future work. First, the current research has only been validated on two modalities of data: ultrasound and fundus color photography. Subsequent studies will require more extensive validation experiments on various other modalities of data. The Discussion emphasizes the theoretical training of SAM and SAM-Med2D on multiple data modalities, suggesting avenues for further exploration in validating SAM’s feature-encoding and -decoding capabilities on additional datasets.

Another crucial point addressed in the Discussion is the scope of validation scenarios, which, thus far, have focused on few-shot, particularly one-shot, scenarios. The challenges associated with handling entirely new data without any labeled samples are recognized, paving the way for future investigations into prompt learning strategies akin to large language models (LLMs) for achieving zero-shot tasks.

Finally, this paper explores several approaches for model enhancement and incorporates knowledge from general model SAM. However, what is still lacking is the comprehensive utilization of data, such as clinically relevant prior knowledge for segmentation targets and addressing the diversity distribution of data. In the future, we can focus on integrating more prior knowledge to ensure segmentation accuracy and optimize through combinations of data augmentation, synthetic data generation, and various generative models.

## 6. Conclusions

This paper introduces AGSAM, a framework leveraging SAM for automatic segmentation with limited training samples. Through an agent-guided model, SAM feature encoding is extracted and fused to generate prompt embeddings for SAM’s mask decoder, eliminating the need for manual inputs. The image-feature-encoding module and its fusion module enhance SAM’s encoding by integrating the guiding model’s capabilities, inheriting SAM’s universal feature representation. The prompt generation module automatically guides mask-decoding predictions, generating sparse to dense prompts to comprehensively constrain and avoid redundancy. The mask decoder, based on SAM, combines decoding experience with prompt embeddings for reliable predictions, integrated with the guiding model’s own predictions. Superior metrics are attributed to these modules effectively extracting and transferring SAM’s pre-trained ‘segmentation experience’ to AGSAM. Additionally, an online feature enhancement training strategy was explored, suppressing previous model feature responses during training to force improved responses to effective features in the next iteration, boosting encoding ability. Together, these ensure AGSAM benefits from varying training data quantities. The proposed method has potential as a new few-shot benchmark leveraging SAM-like pre-training.

## Figures and Tables

**Figure 1 bioengineering-11-00447-f001:**
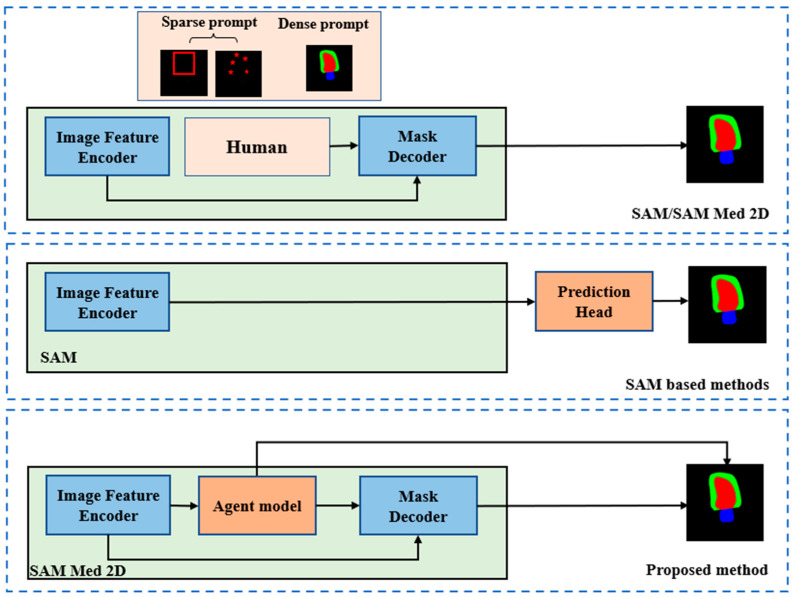
Framework of SAM/SAM-Med2D, SAM-based methods, and proposed work.

**Figure 2 bioengineering-11-00447-f002:**
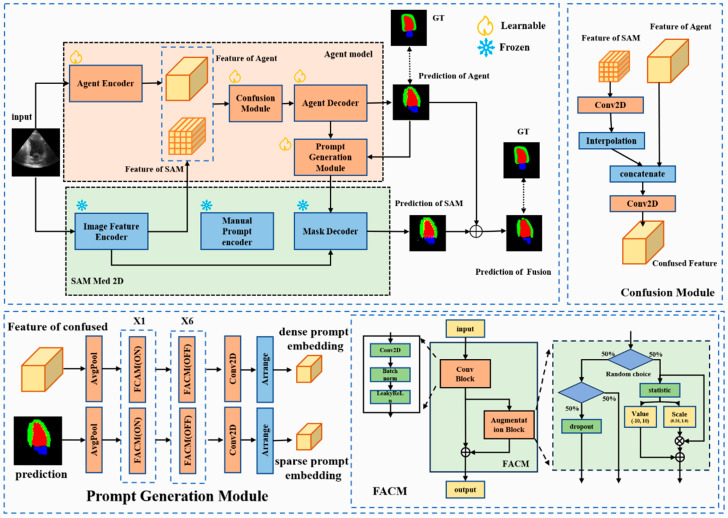
Framework of AGSAM.

**Figure 3 bioengineering-11-00447-f003:**
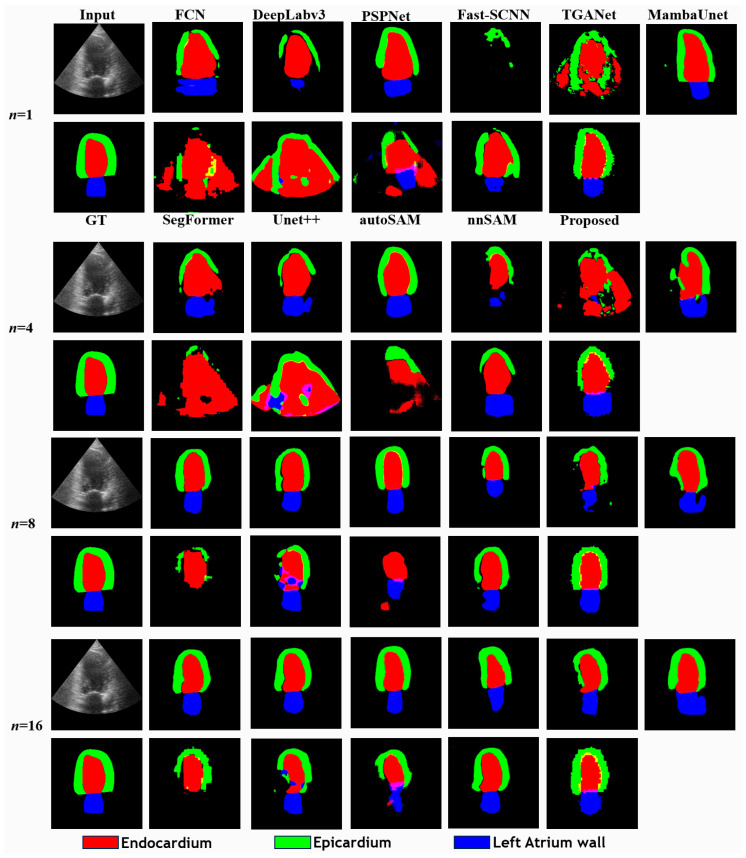
Comparison of predictions of different methods with different training samples (*n*) in CAMUS dataset.

**Figure 4 bioengineering-11-00447-f004:**
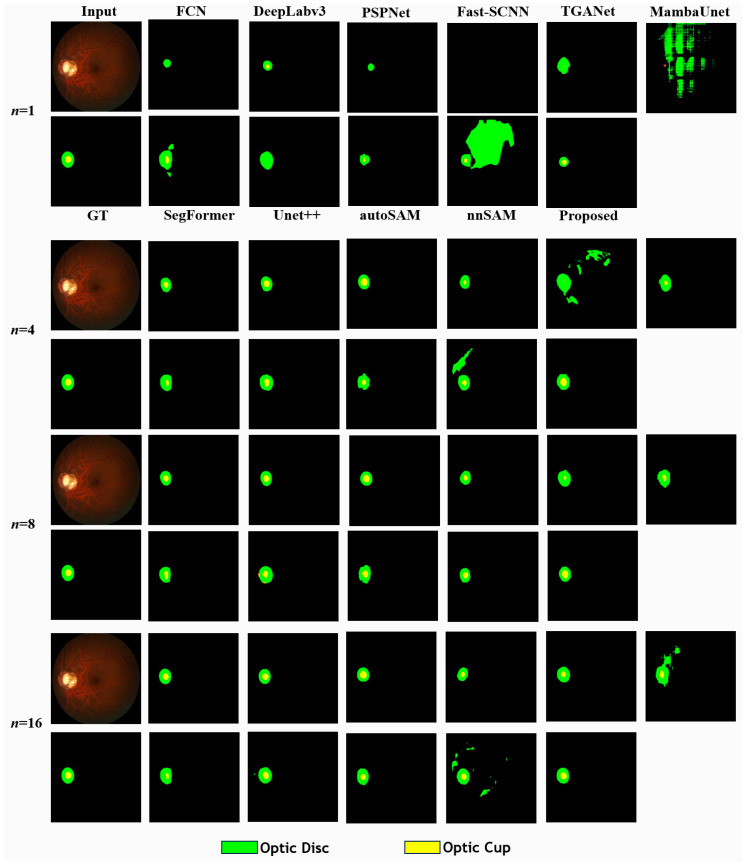
Comparison of predictions of different methods with different training samples (*n*) in REFUGE dataset.

**Figure 5 bioengineering-11-00447-f005:**
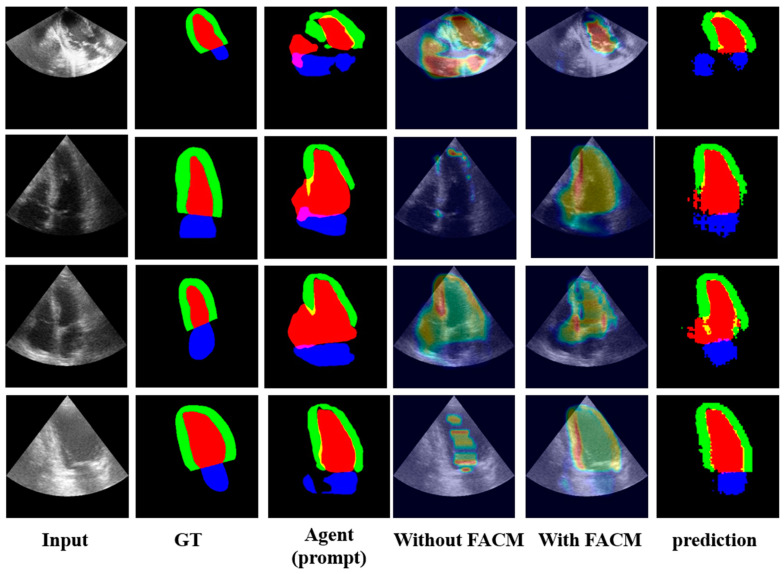
Ablation comparison of predictions with/without FACM.

**Table 1 bioengineering-11-00447-t001:** Comparison results of different methods with different sizes of training sample with DICE and HD in CAMUS. * *p* <  0.05; ** *p* <  0.01; *** *p* <  0.001; ns, not significant (*p* > 0.05).

Method	Metrics															
	DICE								HD							
	Training Sample Size (*n*)															
	1	2	4	6	8	12	16	20	1	2	4	6	8	12	16	20
FCN	0.4819 ***	0.6044 ***	0.5962 ***	0.7507 *	0.7607 ***	0.7954 **	0.8058 ns	0.8036 ns	30.7356 ***	307356 ***	25.0324 ***	12.4143 ns	10.7692 ***	8.7511 *	7.9985 **	8.1526 ns
DeepLabV3	0.5157 ***	0.6312 ***	0.6382 *	0.7417 ***	0.7512 ***	0.7983 ns	0.8029 *	0.8118 ns	26.2979 ***	24.4466 ***	20.1742 ***	12.1354 ns	11.6624 ***	8.3925 ***	8.5411 ***	7.3718 ns
PSPNet	0.5308 ***	0.6198 ***	0.6059 ***	0.7002 ***	0.7206 ***	0.7400 ***	0.7592 ***	0.7681 ***	23.5507 ***	17.9434 ***	18.7494 ***	12.3391 ***	11.1914 ***	11.0718 ***	9.4149 ***	9.7201 ***
Fast-SCNN	0.2311 ***	0.3133 ***	0.4271 ***	0.5263 ***	0.5418 ***	0.6374 ***	0.6328 ***	0.6501 ***	55.0956 ***	36.9229 ***	36.3592 ***	24.4900 ***	23.6173 ***	16.7035 ***	17.6839 ***	17.6589 ***
TGANet	0.3387 ***	0.3503 ***	0.3886 ***	0.6406 ***	0.6435 ***	0.7145 ***	0.7216 ***	0.7069 ***	57.5892 ***	56.7400 ***	44.2247 ***	19.1582 ***	19.0647 ***	14.6703 ***	14.8420 ***	15.9357 ***
SegFormer	0.2637 ***	0.4378 ***	0.2495 ***	0.6084 ***	0.4136 ***	0.6591 ***	0.4737 ***	0.6593 ***	65.4658 ***	50.6095 ***	67.2951 ***	30.3880 ***	47.6450 ***	18.7880 ***	42.6189 ***	19.2131 ***
Unet++	0.2486 ***	0.2915 ***	0.3562 ***	0.6098 ***	0.6779 ***	0.7119 ***	0.7063 ***	0.7405 ***	66.9296 ***	67.5001 ***	68.7728 ***	29.8018 ***	22.7979 ***	19.4106 ***	19.3498 ***	16.7502 ***
autoSAM	0.4482 ***	0.4911 ***	0.4465 ***	0.5844 ***	0.5807 ***	0.6671 ***	0.6642 ***	0.6682 ***	61.9208 ***	51.3325 ***	47.9954 ***	20.5984 ***	20.8936 ***	17.8528 ***	17.9705 ***	18.9544 ***
Mamba-Unet	0.5040 ***	0.5982 ***	0.6089 ***	0.6290 ***	0.6534 ***	0.6528 ***	0.6674 ***	0.7067 ***	23.3985 ***	18.8779 ***	19.3776 ***	18.5073 ***	16.8112 ***	16.9241 ***	15.9930 ***	14.8776 ***
nnSAM (FCN)	0.5087 ***	0.5906 ***	0.5882 ***	0.7564 ns	0.7786 ns	0.8021 ns	0.8069 ***	0.8010 ns	32.6712 ***	31.6343 ***	23.6663 ***	11.1222 *	9.2595 ns	8.1799 ns	7.3214 ***	8.2214 ns
Proposed (FCN)	0.5419 ***	0.6164 ***	0.6103 ***	0.7570 ns	0.7818	0.8060	0.8091	0.8052	25.5238 ***	23.3281 ***	19.4167 ***	10.5465	8.7999	7.9875	7.1839	7.8991
nnSAM (deep)	0.5323 ***	0.6417 ***	0.6435 ns	0.7530 ns	0.7588 ***	0.7915 ***	0.8031 ns	0.8058 ns	24.2617 ***	19.6827 ***	17.5183 ns	12.6661 ns	10.7260 ***	9.3173 ***	8.4043 **	8.0490 ns
Proposed (deep)	0.5758	0.6584	0.6519	0.7599	0.7672 **	0.7973 *	0.8091 ns	0.8104 ns	20.7766	17.5505	16.7683	11.8957 ***	10.3071 ***	8.7514 *	7.8906 *	7.6765 ns

**Table 2 bioengineering-11-00447-t002:** Comparison results of different methods with different size of training sample with DICE and HD in REFUGE. * *p* <  0.05; ** *p* <  0.01; *** *p* <  0.001; ns, not significant (*p* > 0.05).

Method	Metrics															
	DICE								HD							
	Training Sample Size (*n*)															
	1	2	4	6	8	12	16	20	1	2	4	6	8	12	16	20
FCN	0.4790 ***	0.6759 ***	0.7762 ***	0.8210 ***	0.8130 ***	0.8726 ns	0.8718 ns	0.8751 ns	20.0650 ***	10.7355 ***	5.3205 ***	3.0799 **	3.3850 ***	1.7938 ns	1.9800 ns	1.6347 ns
DeepLabV3	0.5657 ***	0.7034 ***	0.8229 ***	0.8145 ***	0.8350 ns	0.8736 ns	0.8754 ns	0.8763 ns	19.1857 ***	8.0980 ***	3.3823 ***	4.2733 ***	2.8859 *	1.7468 ns	1.7494 ns	1.4995 ns
PSPNet	0.2952 ***	0.5397 ***	0.6876 ***	0.7643 ***	0.7935 ***	0.8342 ***	0.8432 ***	0.8349 ***	36.6751 ***	8.7706 ***	6.3812 ***	8.7565 ***	7.6729 ***	5.8643 ***	3.9266 ***	3.8622 ***
Fast-SCNN	0.3473 ***	0.5133 ***	0.6423 ***	0.6507 ***	0.6662 ***	0.7837 ***	0.7746 ***	0.7987 ***	45.4614 ***	31.5551 ***	15.1610 ***	15.2604 ***	15.1658 ***	6.2228 ***	6.7428 ***	6.1936 ***
TGANet	0.5750 ***	0.6537 ***	0.6491 ***	0.6970 ***	0.7471 ***	0.8157 ***	0.8091 ***	0.7976 ***	26.2602 ***	26.0183 ***	31.2509 ***	26.2529 ***	15.8280 ***	8.5710 ***	9.9134 ***	10.0348 ***
SegFormer	0.6014 ***	0.6518 ***	0.7065 ***	0.7574 ***	0.7638 ***	0.7764 ***	0.7559 ***	0.7578 ***	30.1322 ***	16.0105 ***	14.2888 ***	7.9404 ***	8.1005 ***	9.7448 ***	9.4702 ***	9.1201 ***
Unet++	0.5241 ***	0.4020 ***	0.7922 ***	0.8215 ***	0.7961 ***	0.8426 ***	0.8463 ***	0.8563 ***	41.3638 ***	82.2642 ***	9.9493 ***	8.1613 ***	10.7172 ***	10.3751 ***	9.2046 ***	7.2565 ***
autoSAM	0.4723 ***	0.4456 ***	0.7208 ***	0.7627 ***	0.7789 ***	0.8042 ***	0.7972 ***	0.8076 ***	22.5677 ***	35.1402 ***	12.9685 ***	8.2359 ***	8.5600 ***	6.2316 ***	5.2601 ***	5.3802 ***
Mamba-Unet	0.2567 ***	0.3326 ***	0.3225 ***	0.6176 ***	0.6935 ***	0.7320 ***	0.6709 ***	0.7605 ***	41.4738 ***	33.2122 ***	15.8653 ***	13.8365 ***	10.6301 ***	13.8848 ***	16.3959 ***	8.5436 ***
nnSAM (FCN)	0.6049 ***	0.7886 ns	0.7994 ***	0.8172 ***	0.8412 ***	0.8737 ns	0.8668 ***	0.8720 **	13.0598 ***	3.8768 ns	4.5948 ***	3.3072 ***	2.6651 ***	1.7309 ns	1.9841 ns	2.0956 *
Proposed (FCN)	0.7141	0.7898	0.8427	0.8449	0.8432	0.8743	0.8773	0.8800	7.5007	4.6304	2.3837	2.3026	2.3543	1.7615	1.8195	1.5801
nnSAM (deep)	0.6347 ***	0.7007 ***	0.8028 ***	0.8176 ***	0.8395 ns	0.8721 ns	0.8681 **	0.8748 ns	10.5907 ***	9.2379 ***	3.9076 ***	3.6697 ***	2.8225 ns	2.0105 ns	1.9409 ns	1.7629 ns
Proposed (deep)	0.6725 ***	0.7282 ***	0.8075 ***	0.8229 ***	0.8329 *	0.8674 *	0.8741 ns	0.8784 ns	11.8223 ***	7.5185 ***	4.2160 ***	3.3738 ***	2.6646 ns	2.1048 ns	2.2357 ns	1.5954 ns

**Table 3 bioengineering-11-00447-t003:** Results of different AGSAM-based models with different module combinations in ablation analysis.

Ablation study (FCN) *n* = 1	**Modules**			**Metrics**	
	FE	MD	FACM	DICE	HD
	×	×	×	0.4819	33.53
	√	×	×	0.5087	32.67
	√	√	×	0.5324	25.57
	√	√	√	0.5419	25.52
Ablation study (DeeplabV3) *n* = 1	**Modules**			**Metrics**	
	FE	MD	FACM	DICE	HD
	×	×	×	0.5157	26.30
	√	×	×	0.5323	24.26
	√	√	×	0.5689	20.67
	√	√	√	0.5758	20.78
Ablation study (Unet++) *n* = 1	**Modules**			**Metrics**	
	FE	MD	FACM	DICE	HD
	×	×	×	0.2486	66.93
	√	×	×	0.2430	68.49
	√	√	×	0.2592	66.88
	√	√	√	0.2615	66.46

**Table 4 bioengineering-11-00447-t004:** Results of AGSAM with different fusion weight ratio of agent and SAM.

Fusion Weight of Agent and SAM			Training Sample Size (*n*)			
Agent	SAM	Metrics	1	4	8	16
0.1	0.9	DICE	0.5758	0.6428	0.7683	0.7986
0.25	0.75	DICE	0.5683	0.6427	0.7672	0.8091
0.5	0.5	DICE	0.5581	0.6385	0.7618	0.8063
0.75	0.25	DICE	0.5536	0.6357	0.7591	0.8023
0.9	0.1	DICE	0.5518	0.6345	0.7580	0.8005
0.1	0.9	HD	20.7766	18.2607	10.1163	8.0842
0.25	0.75	HD	20.6299	18.4923	10.3071	7.8906
0.5	0.5	HD	20.5974	18.9707	10.6805	9.6143
0.75	0.25	HD	20.6733	19.2346	10.8251	11.4226
0.9	0.1	HD	20.6801	19.3462	10.8708	12.2420

## Data Availability

Data sources are cited within the article (CAMUS dataset: https://www.creatis.insa-lyon.fr/Challenge/camus/index.html accessed on 24 March 2024 and REFUGE dataset: https://refuge.grand-challenge.org/ accessed on 24 March 2024). Our source code will be made available on request.

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
