# Peer review of "AGSAM: Agent-Guided Segment Anything Model for Automatic Segmentation in Few-Shot Scenarios"

_bioengineering, 2024, doi:10.3390/bioengineering11050447_

Round 1

Reviewer 1 Report

Comments and Suggestions for Authors

1. The paper claims that AGSAM outperforms other methods but does not provide a detailed comparison with current state-of-the-art approaches in medical image segmentation. Including such comparisons would enhance the credibility of the claims.

2. The description of the FACM is vague and lacks technical depth. Providing more details on its architecture, how it integrates with the overall system, and why it improves feature stability would be beneficial.

3. While the paper targets few-shot scenarios, it does not adequately demonstrate AGSAM's performance in these conditions. More comprehensive testing with varying degrees of data scarcity would better validate its effectiveness.

4. The paper predominantly focuses on cardiac ultrasound segmentation. It's unclear how well AGSAM generalizes to other types of medical imaging. Including results for additional imaging modalities would provide a more comprehensive evaluation of the method's adaptability.

5. While automating prompt generation is a notable achievement, the paper should also address potential limitations or inaccuracies introduced by fully automated methods. Discussing how AGSAM ensures the quality and relevance of generated prompts would add depth to the discussion.

6. Author should investigate more background study to understand the gaps and state of the art methods in this domain like this "https://doi.org/10.1016/j.compbiomed.2023.106646", "https://doi.org/10.1109/ACCESS.2023.3244952".

7. The paper claims superior performance but does not provide detailed statistical analysis to support these claims. Including metrics like sensitivity, specificity, and area under the curve (AUC), along with p-values for statistical significance, would strengthen the argument.

8. There is no mention of the computational requirements or efficiency of AGSAM compared to other methods. Given the importance of computational resources in clinical settings, discussing the model's efficiency and scalability would be relevant.

9. The paper does not discuss how the dataset's diversity (or lack thereof) might affect the generalizability of the results. Including a discussion on potential biases in the training data and how they were mitigated would enhance the paper's thoroughness and relevance to real-world applications.

Comments on the Quality of English Language

Extensive editing of English language required

Reviewer 2 Report

Comments and Suggestions for Authors

Can author elaborately discuss how few-shot learning enables pre-trained model in the context of AGSAM and why the proposed model called the novel one over the existing AutoSAM scheme? What is the computational complexity & the experimental error of data pre-processing in the FACM in AGSAM? Present the parameters with values that have used in the experimental analysis in a tabular format under section 4.
